# The Italian Pathway for Energy Transition: From the Coal Phase Out to the Problems Related to Natural Gas

**Claudia Cafaro \*, Paolo Ceci \***  **and Antonio Fardelli**

Institute of Atmospheric Pollution Research, Division of Rome, C/O Ministry for Ecological Transition, Via Cristoforo Colombo 44, 00147 Rome, Italy

\* Correspondence: cafaro@iia.cnr.it (C.C.); ceci@iia.cnr.it (P.C.); Tel.: +39-06-57225018 (C.C.); +39-06-57225965 (P.C.)

**Abstract:** In Italy, the de-carbonization process descending from the National Energy Strategy (NES) of the November 2017 and the National Energy and Climate Plans (NECPs) of December 2019 has led two specific effects: the progressive ending of coal use as a fuel for the production of electricity and the emanation of authorization acts for the conversion to natural gas of existing coal-fired power plants and/or for the construction of new power plants powered by natural gas. These new plants will be technologically advanced and designed to guarantee the safety of the national electricity grid in periods of greatest energy demand and will have to support the growing part of energy produced with plants powered by renewable energy sources (especially wind and photovoltaic). This reference context will necessarily have to take into account the recent gas supply difficulties due to the Russian-Ukrainian conflict, which could lead to a reconsideration of the projects for the construction of new gas plants by focusing on other energy sources. This paper hence aims to quantify and to evaluate the effective environmental benefits for atmospheric emissions, related to the replacement of coal with natural gas. Starting from the electrical powers replaced and installed, the potential reductions in greenhouse gas emissions will be examined, comparing the current emission situation in terms of $CO_2$, with the future scenarios deriving from the construction of thermoelectric plants whose projects are currently under authorization.

**Keywords:** de-carbonization; energy generation; $CO_2$ emissions; power plants

## 1. Introduction

In 2020, the overall global emissions of greenhouse gases expressed as $CO_2$ were equal to 34,807 Mt; therefore, the international policies implemented by the main industrialized countries must address the issue of climate change, which require all states to implement adequate strategies for achieving the goals set for 2030 by the Paris Agreement [1] and for 2050 by the report of the Intergovernmental Panel on Climate Change (IPCC) [2]. To pursue these objectives, many countries are adopting low-carbon solutions, with reference to the sectors that are the major contributors to climate-changing emissions, such as the energy sector. In this context, many nations are implementing an important transformation of the energy sector, replacing gradually the most polluting fuels with lower environmental impact systems. In countries characterized by an extensive use of polluting fuels, such as coal, this transformation could generate worrying socio-economic problems. For example, in China, the gradual replacement of coal with other types of energy has a negative impact on approximately 4 million people working in fossil fuel mines and on a greater number of workers involved in supporting fuel supply chain; therefore, energy transition actions should include strategies to protect the economic and social welfare [3]. All decarbonization objectives, defined at a global level, must necessarily balance the aspects related to the achievement of significant environmental goals with those related to humans needs and social stability.

At the European level for over 20 years, through the application of laws on integrated pollution prevention and control, a process has been undertaken which has allowed the gradual reduction of emissions from the main industrial installations. The current European directive for this sector, the Industrial Emissions Directive (IED) [4] provides that industrial plants are comply with the specific emission limit values fixed by the Best Available Techniques (BAT). For the energy sector, the adoption of the BATs concerns thermoelectric plants with a thermal power more than 50 MW (so called Large Combustion Plants—LCP) as defined by the specific European implementation decision [5]. The result of the application of the obligations deriving from the IED is that large combustion plants now emit air pollutants seven times less than they did 20 years ago [6].

The IED directive is mainly addressed to regulate the substances associated with the pollution generated directly by specific industrial installation at a local level but does not take into account the greenhouse gas emissions into the atmosphere. This type of emissions is regulated by the 2003/87/EC directive that setting up the greenhouse gas emissions trading system at European level in order to facilitate their reduction [7].

The energy sector is one of the major sources responsible for atmospheric emissions of greenhouse gases; for this reason, many of the interventions planned at an international level to limit progressively the contribution to global climate-altering gas emissions over the next few years are aimed at transforming energy production. European production activities produce over 10 per cent of direct global greenhouse gas emissions [8], the largest contribution of this percentage is caused by the energy sector [9] (see Figure 1). To tackle this result, long-term European policies provide for reach a level zero of net greenhouse gas emissions into the atmosphere by 2050 (see Figure 2) [10].

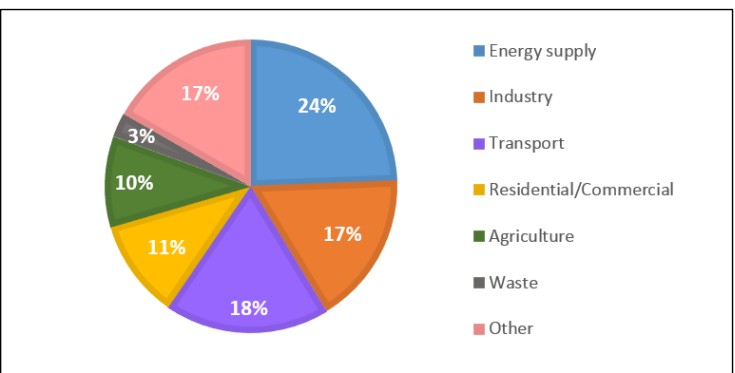

**Figure 1.** Percentage of European greenhouse gas emissions by aggregated sector in the 2017 [9].

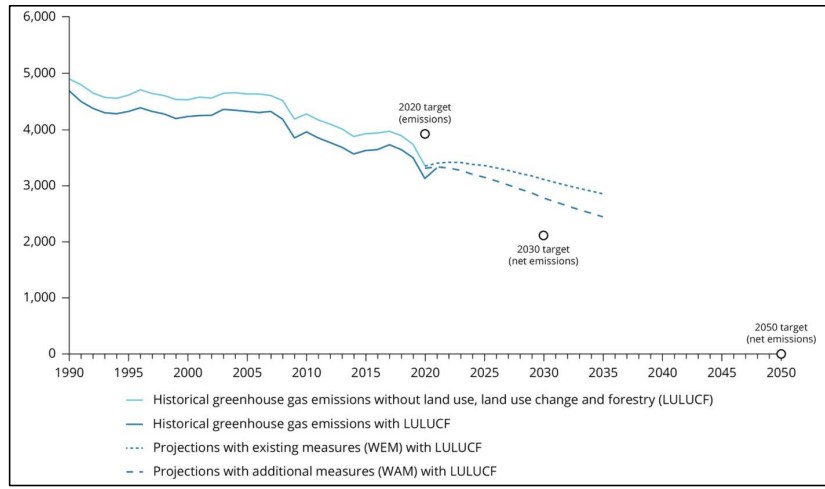

**Figure 2.** The greenhouse gas emission targets, historical and future trends for the EU Member States (EU-27) indicated as million tons of $CO_2$ equivalent (Mt $CO_2$ eq.) [11].

Figure 2 shows how the EU has largely reached and overcome the target of reducing greenhouse gas emissions by 20% fixed for 2020. This data is certainly partly influenced by the measures adopted to combat the spread of COVID-19, which in 2020 led to a slowdown in economic activities and transport with evident environmental benefits. Nevertheless, is also the result of the start of the processes for the reduction and replacement in the use of fossil fuels undertaken by European countries, with incentives for energy production from renewable and less polluting sources. The projection hypothesized for 2030, on the other hand, does not seem to allow the achievement of the further reduction in greenhouse gas emissions expected, which will have to reach a percentage higher than 40%, therefore relevant and effective interventions will be necessary, such as to reach a rapid and concrete decrease of the greenhouse gases in the atmosphere [11].

The goal of a total de-carbonization in energy production can only achieved through an important transformation of the whole energy sector, under the progressive and definitive dropout of fossil fuels, encouraging and intensifying the use of renewable energy sources and creating networks of safe and efficient energy supply, transfer, and storage.

To achieve these goals, the EU countries by the end of 2019 presented the National Energy and Climate Plans (NECPs) to identify the actions to be implemented over a period of 10 years (2021–2030) to define a pathway to 2030 targets. The different national plans must establish the minimum conditions for implementing an effective energy transition, covering the following issues:

- Greenhouse gas emission reductions;
- Renewable energy;
- Energy efficiency;
- Electricity interconnections;
- Research and innovation.

Therefore, each country has prepared a document outlining policies and sectors of intervention to define the path for the collective achievement of the 2030 objectives (see Table 1) [12].

**Table 1.** European 2030 targets [12].

| 2030 Targets | | | | |
|---|---|---|---|---|
| Greenhouse Gas Emissions | Renewable Energy | Energy Efficiency | Interconnection | Climate in EU-Funded Programs |
| $\leq -40\%$ | $\geq 32\%$ | $\geq 32.5\%$ | 15% | 2021–2027 25% |

## 2. Methodology

This paper intends to outline and analyze the de-carbonization process undertaken in Italy. Starting from the planning documents that identify the policies for achieving the medium and long-term objectives, a specific analysis will be conducted approximately the targets indicated and the benefits expected from the application of these measures.

The Italian National Energy and Climate Plan provides an ambitious de-carbonization scenario with total phase out of coal power generation by 2025. This objective needs to replace the energy produced by the current coal-fired thermoelectric plants, or with the conversion of these plants to natural gas or with new installations technologically advanced also fueled by natural gas, or with plants fueled by renewable energy [13].

Figure 3 highlights how in Italy, compared to the current situation, after 2025 the contribution of energy produced with solid and liquid fuels will be eliminated. Considering the eight coal-fired power plants still in operation, six presented natural gas conversion projects; the related environmental assessment and authorization procedures are underway at the Ministry for the ecological transition. Of the two remaining plants, one works with a multi-fuel group that can already be fueled with natural gas instead of coal, the other is located in Sardinia, a region that is not yet reached by methane gas. A specific gas conver-

sion project under evaluation, was also presented for the only plant still operating with exclusive heavy fuel oil supply. Therefore, almost the total capacity currently authorized using coal or oil as fuels, will probably be authorized to work with natural gas, ensuring the maintenance of a large part of the installed capacity.

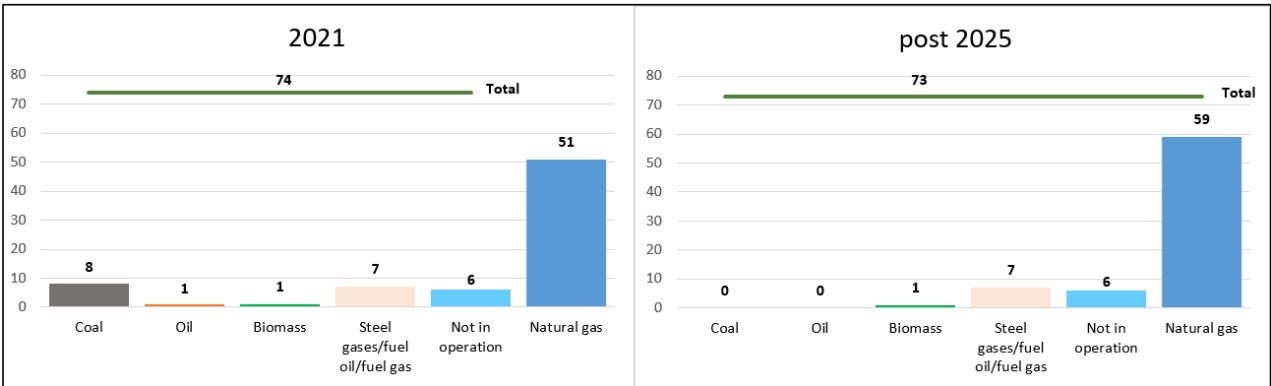

**Figure 3.** Distribution by fuel of existing Italian power plants with capacity ≥300 MWth in 2021 and post phase out of coal [14].

Based on the elements above represented, this work analyses the present emission framework in terms of greenhouse gas emissions, compared with the possible future scenario where solid and liquid fuels will be largely replaced by gas. This is in order to evaluate the real environmental benefits obtained and to define the contribution of natural gas to the de-carbonization process.

## 3. The Italian Reference Context

The total Italian emissions of greenhouse gases, expressed as tons of $CO_2$ equivalent, are decreased from approximately 519,551 kt in 1990 to 389,217 kt in 2019, with a reduction of approximately 25%. This reduction is coherent with the European situation, where greenhouse gas emissions are decreased from 5,538,761 kt in 1990 to 3,985,673 kt in 2019, corresponding to a reduction of approximately 28% [15]. Figure 4 shows the reduction trend from 1990 to 2019 for the European Union and for Italy; the decrease in greenhouse gas emissions in Italy has followed the same trend as in Europe. This important result is mainly due to the actions taken in recent years to improve environmental performance in the energy and industrial sectors, preferring less polluting fuels for energy production and encouraging the use of energy produced from renewable sources [16].

In order to achieve the greenhouse gas emission reduction targets, identified at European level, the Italian National Energy and Climate Plan has defined a path that provides for the total phase out of the production of electricity obtained from the combustion of coal by 2025. The plan establishes to replace the energy obtained from coal with energy produced by a mix based on a growing and significant portion of energy produced by renewable sources and by a remaining fraction obtained from gas plants. Currently, coal-fired generation in Italy consists of eight operating power plants, contributing to eight GW electrical generation capacity (Figure 5). For coal-fired power plants, considering their stop of activity by the end of 2025, the gradual process of definitive shutdown of the groups has begun. Table 2 shows the thermal and electrical capacities for the different production units of the plants in operation; for some units, the closure took place earlier than the deadline set for December 2025.

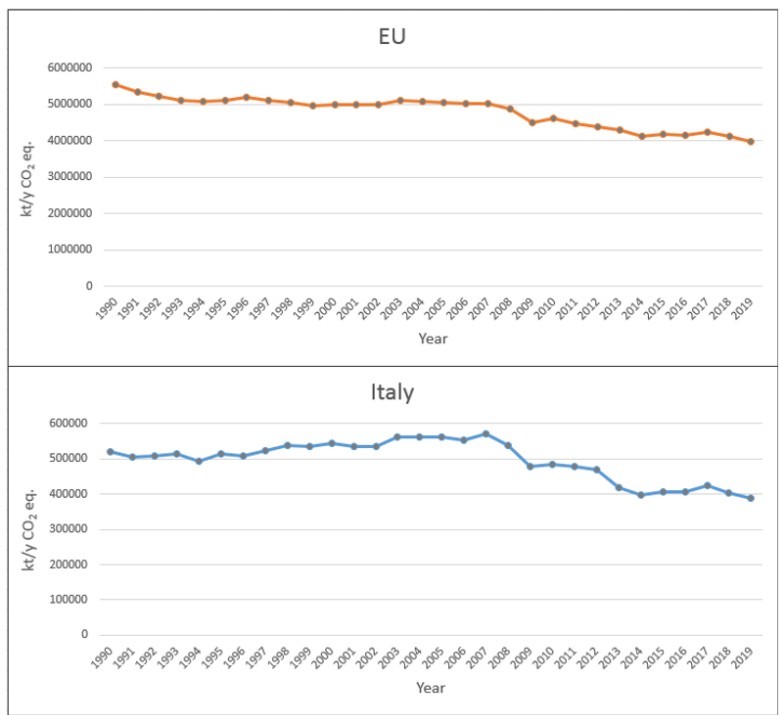

**Figure 4.** The greenhouse gas emissions trend from 1990 to 2019 in EU and in Italy indicated as kilo tons of $CO_2$ equivalent (kt $CO_2$ eq.) [15].

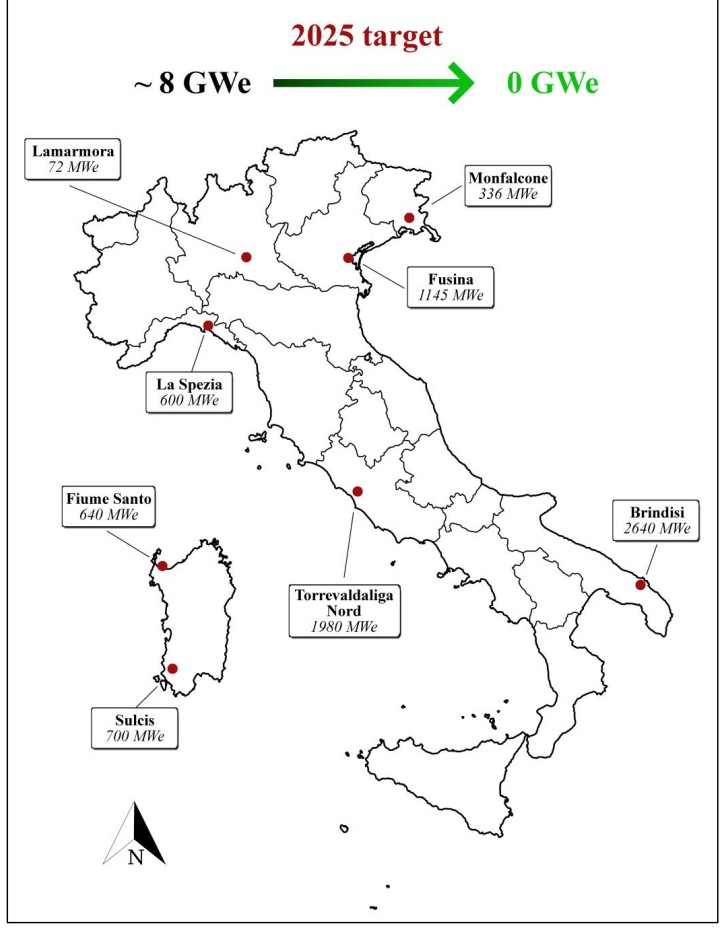

**Figure 5.** Italian coal fired combustion plants.

**Table 2.** Italian operating coal fired combustion plants [14].

| Coal-Fired Combustion Plant | Power Generation Unit | | | |
| --- | --- | --- | --- | --- |
| | Unit | Thermal Capacity | Electrical Capacity | Status |
| Torrevaldaliga Nord Civitavecchia (RM) | TN2 | 1420 MWth | 660 MWe | Operating |
| | TN3 | 1420 MWth | 660 MWe | Operating |
| | TN4 | 1420 MWth | 660 MWe | Operating |
| Brindisi | GR1 | 1640 MWth | 660 MWe | Operating |
| | GR2 | 1640 MWth | 660 MWe | Closed (1/01/2021) |
| | GR3 | 1640 MWth | 660 MWe | Operating |
| | GR4 | 1640 MWth | 660 MWe | Operating |
| Monfalcone (GO) | GR1 | 418 MWth | 165 MWe | Operating |
| | GR2 | 433 MWth | 171 MWe | Operating |
| Fusina Porto Marghera (Venice) | GR1 | 415.2 MWth | 175 MWe | Closed (31/12/2021) |
| | GR2 | 430.8 MWth | 190 MWe | Closed (31/12/2021) |
| | GR3 co-combustion coal/CSS | 792.8 MWth | 390 MWe | Operating |
| | GR4 co-combustion coal/CSS | 792.8 MWth | 390 MWe | Operating |
| Lamarmora (Brescia) | GR3 | 200 MWth | 72 MWe | Operating with gas (Phase out coal at 31/12/2020) |
| La Spezia | SP3 | 1540 MWth | 600 MWe | Closed (31/12/2021) |
| Sulcis Portoscuso (SU) | SU2 | 800 MWth | 410 MWe | Operating |
| | SU3 | 670 MWth | 290 MWe | Operating |
| Fiume Santo (SS) | GR3 | 800 MWth | 320 MWe | Operating |
| | GR4 | 800 MWth | 320 MWe | Operating |
| Total | | 18,912.6 MWth | 8118 MWe | |

The Italian NECPs requires that the de-carbonization process is realized through an acceleration in the implementation and use of renewable energy. This process must be accompanied by the implementation of suitable storage systems for approximately 3 GW electrical generation capacity, necessary to guarantee the safety of the grid in consideration of a discontinuous production of energy from renewable sources, and through the additional availability of gas plants, for approximately 3 GW electrical generation capacity, of which approximately 50% related to the phase out of coal. This new gas capacity will have to offset the portion of energy currently produced with coal, ensuring the adequacy of the national electricity system even in periods of greatest demand. For this purpose, the new gas plants will be designed to come into operation quickly in case of need and to respond to wide variability in the electrical system.

In Italy, with the ministerial decree of 28 June 2019, a tool was defined to promote investments in projects for the construction of new gas power plants and storage systems, especially to support energy production from renewable sources [17]. This system, named capacity market, provides specific incentives related to the availability of energy plants to start operating both in the medium-long term and in the short term, especially in periods

of increased energy demand linked to seasonal or climatic factors. The purpose of the capacity market is to support the construction of production plants (programmable and non-programmable) and storage systems by awarding segments of electricity capacity assigned through specific auctions. In 2019, the first two auctions will be referred to the allocation of a new capacity to make available in the years 2022 and 2023. In these auctions, a total new capacity of 5.8 GW of electricity was assigned (1.8 GW for the 2022, and 4 GW for 2023); the third auction was held in February 2022, for the distribution of a new capacity of 3.8 GW to make available in the year 2024 (Table 3). The owners awardees of the auctions are obliged to make available the allocated capacity for which they receive a specific remuneration, starting by the referring date [18].

**Table 3.** New Italian electricity generation capacity [18].

| 2025 Coal Phase Out | 2019 Auctions | | 2022 Auction |
|---|---|---|---|
| | Granted New Capacity for 2022 | Granted New Capacity for 2023 | Granted New Capacity for 2024 |
| −8 GW | 1.8 GW | 4 GW | 3.8 GW |

This tool will allow the replacement and conversion of the current coal plants (inefficient and highly polluting) with the latest generation of gas plants; plants that are technologically advanced, flexible, and environmentally more sustainable. Their realization will ensure the overcoming of the critical issues related to the progressive replacement of conventional programmable sources with energy from non-programmable and discontinuous renewable sources and the reduction of electricity availability associated with the decarbonization objectives.

With reference to the Italian electricity system, must be considered that the available electricity capacity is lower than that installed, this condition is due, among other things, to discontinuous production from renewable sources or to failures and maintenance of plants and the grid. To evaluate the adequacy of the electricity system, the LOLE factor (Loss Of Load Expectation) is used as an indicator, which indicates the number of hours per year in which it is likely that the energy demand exceeds the available quantity. An electrical system is considered adequate if the LOLE factor does not exceed 3.

The construction of new electrical production plants leads to an increase in the adequacy of the system, with a distinction between programmable sources and non-programmable sources. Due to its continuous energy production characteristics, a traditional thermoelectric plant will allow a greater increase in the adequacy than a plant that uses renewable and discontinuous energy sources. Therefore, the growth of the percent of energy produced with renewable sources, envisaged by the de-carbonization objectives, determines a possible criticality of the system with an increase in the LOLE factor. The 2021 report on the adequacy of the Italian electricity system prepared by TERNA (the Italian operator responsible for managing the national electricity grid) highlights that the scenario analyzed in 2025 does not comply with the adequacy criterion with LOLE equal to 3 h/year. The report shows that the stop to 2025 of electricity production by coal is not currently compensated by the present implementation of renewable sources, storage systems and the new gas capacity assigned with the 2019 auctions. Only with the complete realization of the projects assigned to the auctions of 2019 and with the addition of further capacity, with the coming auctions foreseen by the capacity market, it will be possible to reach again adequate safety standards. To have an adequate electricity system, Italy must be able to count on an electricity generation capacity of at least 54 GW, a reduction of this value, associated with the interruption of the operation of coal and fuel oil plants, involves the increase of the LOLE factor [19].

Particularly critical is the situation of Sardinia because this island is not currently reached by methane gas, therefore the electricity necessary to cover internal needs is now largely ensured by the operation of two coal plants, placed one in the north and one in

the south of the island. A portion of the energy supply can also take place through a submarine cable interconnection that connects Sardinia with Lazio. The 2021 adequacy report published by TERNA shows that the closure of the two coal plants would expose the island to a highly critical situation. Criticality that could only be overcome with the construction of a new 500 MW electrical capacity distributed on the island (at least 200 MW in the north, and 300 MW in the south of Sardinia) and with the construction of a new electricity connection with the Italian peninsula; TERNA's report concludes believing that in Sardinia it will be possible to stop the coal-fired power plants only after the complete construction of the new connection (expected entry into operation between 2026 and 2028) and the new electricity availability [19].

## 4. Discussion

From the data obtained by the annual reports draw up by TERNA, in Italy in 2020 there were 4884 thermoelectric plants including geothermal plants and thermoelectric plants powered by renewable sources (Bioenergies), for a total of 5915 units and for an installed nominal power of almost 60 GWe [20]. This capacity is distributed throughout the entire peninsula including islands, with significant availability especially in the more industrialized northern regions (Table 4). These data were compared with those extracted from the website of the Ministry of ecological transition, because the Italian Ministry of ecological transition is the competent authority with regard to thermal power plants with ≥300 MW capacity. The Ministry issues the permits that prescribe the operating environmental conditions in line with the best available techniques defined at European level for these plants [14]. This type of power plants is just over 70 with a total installed capacity of approximately 53.4 GWe, which compared to 59.6 GWe, inclusive of all plants in operation in 2020, represents a percentage of approximately 90%. Therefore, it can be considered that the higher power plants make up almost the entirety of the Italian thermoelectric sector.

**Table 4.** Distribution of power plants in Italy in 2020 [20].

| Region | Number of Plants | Number of Sections | Capacity (MWe) |
|---|---|---|---|
| Piemonte | 458 | 527 | 4580.3 |
| Valle d'Aosta | 16 | 18 | 11.5 |
| Lombardia | 1174 | 1362 | 11,463.7 |
| Trentino Alto Adige | 344 | 411 | 225.2 |
| Veneto | 525 | 593 | 2845.1 |
| Friuli Venezia Giulia | 181 | 210 | 1586.4 |
| Liguria | 35 | 57 | 1415.4 |
| Emilia Romagna | 906 | 1108 | 6091.2 |
| **Total Northern Italy** | 3639 | 4286 | 28,218.9 |
| Toscana | 270 | 321 | 2938.1 |
| of which geothermal | 34 | 36 | 918.8 |
| Umbria | 96 | 113 | 534.5 |
| Marche | 122 | 132 | 538.7 |
| Lazio | 188 | 259 | 5311.2 |
| **Total Central Italy** | 676 | 825 | 9322.5 |
| Abruzzo | 51 | 68 | 1414.2 |
| Molise | 21 | 29 | 1106.4 |
| Campania | 155 | 184 | 1895.1 |
| Puglia | 95 | 123 | 6151.1 |
| Basilicata | 45 | 58 | 122.4 |
| Calabria | 67 | 81 | 3823.4 |
| Sicilia | 92 | 204 | 5343.3 |
| Sardegna | 43 | 57 | 2221.4 |
| **Total Southern and Insular Italy** | 569 | 804 | 22,077.3 |
| **Total** | 4884 | 5915 | 59,618.7 |

Taking as a reference the thermoelectric power plants with ≥300 MWth capacity and processing the data referring to the hours of operation, it results that a very few plants work at full capacity throughout the year. In 2019, chosen as the reference year as it was not influenced by the effects that the spread of COVID-19 caused in 2020, with significant consequences also in the trend of energy demand, the operating hours data shows that on average the Italian thermoelectric plants worked for approximately 4100 h in a year, less than half of the total hours available in a year of 8760.

Processing the detailed data of all the installations in operation it also appears that only 9 plants have worked for a number of hours exceeding 7000 and, between these, only 2 for more than 8000 h. Table 5 shows the aggregate average data of the operation hours of the plants distributed by geographic area; this data shows how the reduced operation of existing plants is homogeneous and generalized throughout Italy and does not depend on situations related to the territorial context.

**Table 5.** Operation hours of power plants in Italy in 2019.

| Geographic Area | Operation Hours |
|---|---|
| Northern Italy | 4053 |
| Central Italy | 4129 |
| Southern and Insular Italy | 4118 |
| Average | 4100 |

To complete the analysis of the Italian electricity sector must be included also the sector of plants using renewable sources, a sector that in recent years has been continuously growing, also in accord with the de-carbonization targets. For these plants in 2020 an installed capacity of 56.59 GW was reached, which represents almost half of the total capacity present in Italy, related to approximately 950,000 plants. This installed capacity is mainly attributable to photovoltaic plants with 21.65 GW, wind farms with 10.91 GW and water plants with 19.11 GW [20].

Compared to the situation illustrated, in the next few years there will be a deep transformation of the Italian electricity sector with the closure of coal plants and the increase of gas plants, plants powered by renewable energy sources and storage systems. This transformation is in line with the provisions of the regulatory and planning instruments aimed at reducing greenhouse gas emissions and it is in accordance with the results of the procedures carried out in the context of the capacity market.

This future arrangement will necessarily consider the recent difficulties in the supply of gas deriving from the Russian-Ukrainian conflict, which led to the Italian government to looking for alternative solutions for the supply of methane, limiting as much as possible the dependence on gas coming from Russia. This new situation, that occurred during 2022, could represent a risk approximately the energy objectives envisaged, especially as regards to the projects for the construction of new gas plants, because may be limitations in the supply of fuel and consequently in the operation of the plants. To overcome this criticality a plan has been drawn up at European level, to limit the dependence of Europe on Russian gas, maintaining the established emission reduction targets [21].

In Italy, some urgent measures to be implemented quickly, have been set taking into account that in 2021 the gas coming from Russia represented approximately 40% of the national demand, with 29 billion cubic meters out of 76 billion cubic meters of gas consumed. The energy demand in terms of gross energy availability in Italy for 2021 was represented for 40.9% by natural gas, for 32.9% by oil, for 19.5% by renewable energy sources, for 3.6% from solid fuels [22]. It was therefore necessary to urgently issue a plan on energy policies, to diversify the sources of gas supply, to guarantee the security of the national grid.

For dealing with these issues, the Italian Government in July 2022 has provided special actions in a specific document called "National plan for the containment of natural gas consumption". The gas plan, without prejudice to the national de-carbonization programs

of the national energy system, envisage measures to diversify the source of the imported gas, mainly using floating structures, which are more flexible and with shorter construction times than fixed structures. The increase in national regasification capacity will therefore be achieved by resorting to Floating Storage and Regasification Units (FSRU) to connect to the existing pipe network. These units are believed strategic, essential and urgent considering the current strong dependence on Russian gas [23].

In this regard, the necessary authorizations are being acquired for the use of two FSRUs with a capacity of approximately 5 billion cubic meters each. The ships will be placed in strategic positions, one in the port of Piombino in Tuscany overlooking the Tyrrhenian Sea and one in the Adriatic Sea in front of the city of Ravenna. The final objective of the initiatives undertaken by the Italian government is to replace approximately 30 billion cubic meters of gas from Russia with approximately 25 billion cubic meters of gas from different sources by 2025, filling the residual difference with the use of renewable sources and implementing energy efficiency improvement policies [24]. The quantities of gas obtained must be sufficient also to ensure the operation of existing and future thermoelectric plants, which will be powered exclusively by methane gas.

About the Italian energy sector, considering that interventions planned by the Italian government allow to maintain unchanged the expected transformation process, an exam was conducted on the submissions presented to the Ministry of ecological transition, to obtain the necessary authorizations for the construction of new plants or for the revamping of existing plants. For this analysis, was chosen as a reference the last three-year period (2019–2021). This choice was made considering that the period is consistent with the timing of implementation of the programmatic guidelines proposed by the Italian NECP (draw up in 2019 and published in its final version in January 2020), but also whit the timing dictated by the first two capacity market auctions of November 2019. In detail, the first auction considers the realization of the projects by 2022 and the second auction by 2023.

Table 6 shows the number of projects presented in implementation of the decarbonization and energy transformation objectives envisaged. Consistently with this pathway, many projects are referring to plants that use renewable energies, especially photovoltaic plants, and wind farms. In accordance with this framework, various projects for energy storage systems have also been presented; necessary to be able to store the energy surplus produced at certain times of the year by renewable energy plants, to be reused in periods in which the availability of these energy sources decreases. With reference to gas-fired thermoelectric plants, there is a considerable number of projects which include both proposals for new plants and revamping of existing plants, with replacement or addition of electricity production units able to respond quickly to the demands of the electricity market. In fact, with the significant increase of the plants that use renewable energy sources, traditional power plants must adapt, seeking technologies that allow rapid start-up times and that are suitable for frequent start-up and shut down situations, necessary to compensate the lack in production of energy from discontinuous sources. These plants so-called "peaker systems", are built to cope with peaks in the demand for energy, to guarantee the safety of the electricity grid always.

**Table 6.** Projects presented from 2019 to 2021 at the Ministry of ecological transition [14].

| Type of Plants | Number of Projects |
|---|---|
| Power plants | 71 |
| New plants and revamping of existing plants | 45 |
| Energy storage facilities | 22 |
| Renewable sources | 488 |
| Photovoltaic | 245 |
| Eolic on-shore | 221 |
| Eolic off-shore | 11 |
| Hydroelectric | 11 |

For the projects of new gas-fired thermoelectric plants or for the revamping of existing plants, the comparison between the currently installed capacity and the future one, obtainable from the realization of the projects presented, leads to an increase of approximately 3.22 GW in the nominal electrical generation capacity. This means that, despite the significant and growing use of renewable energy sources, the pursuit of decarbonization objectives, based on the proposals made, will lead to an increase in the installed capacity of plants that use fossil fuels. Consequently, it could occur that the reduction of climate-altering gas emissions, deriving from the transition from coal to a less polluting fuel such as methane gas, would be nullified by the increase in the number and capacity of thermoelectric units that will come into operation in the coming years.

To estimate the potential emission impacts of the "future" layout of the thermoelectric production sector, based on the data collected and processed, the quantity of $CO_2$ emitted by the Italian plants in different temporal and production conditions was calculated using an algorithm draw up by the authors and following the methodological approach represented in the flowchart shown in the Figure 6.

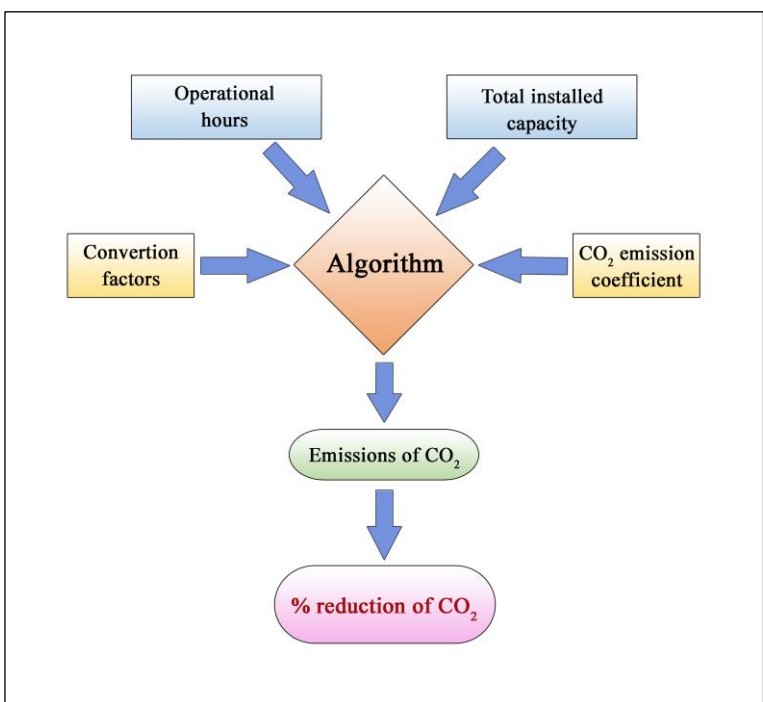

**Figure 6.** Conceptual flowchart representing the methodological approach adopted.

In particular, the theoretical data relating to the tons of $CO_2$ emitted in 2019 by the plants in operation were processed, based on the average hours worked (approximately 4100 h/year) and considering the different types of fuel used; these calculations were also carried out at maximum theoretical operation time (8760 h/year). The results obtained were compared with similar calculations developed in the conditions supposed as post 2025 (use of natural gas as the only fuel, and new production layout resulting from the construction of the plants envisaged as part of the Capacity Market).

The following algorithm was used for the processing:

$$\text{Mt di CO}_2 = \sum (\text{MW}_{th} \times h \times C_{J/W} \times \text{CEF}_x), \tag{1}$$

$\text{MW}_{th}$: thermal combustion capacity; h: hours of operation; $C_{J/W}$: conversion factor from MWth to MJ; $\text{CEF}_x$: $CO_2$ emission coefficient expressed in tons emitted per TJ produced; characteristic data for each fuel (Coal, Fuel Oil, Natural Gas).

For the calculations were considered the following parameters, extrapolated from literature data referring to 2019 [16]:

$C_{J/W}$ = 0.0036 TJ/MWth;
$CEF_{coal}$ = 95,278 t $CO_2$/TJ [16] (p. 477);
$CEF_{Fuel\ Oil}$ = 76,594 t $CO_2$/TJ [16] (p. 475);
$CEF_{Natural\ Gas}$ = 57,632 t $CO_2$/TJ [16] (p. 473).

From the analysis of the results reported in Table 7, emerges that the effective reduction of $CO_2$ emissions after the complete replacement of solid and liquid fossil fuels with natural gas, is limited to 3.50%, if referring to the average operating hours recorded in 2019 (4100 h/a), and to 3.48%, considering the maximum possible operation time of the plants (8760 h/a) (Figure 7). Evaluations were carried out considering the emission factors of 2019 and projecting them also in the post-2025 layout.

**Table 7.** Scenarios of Italian $CO_2$ emission from thermoelectric sector, ante and post de-carbonization.

| Fuel | MW$_{th}$ | Hours | Year | Mt di $CO_2$ [1] |
|---|---|---|---|---|
| Carbon<br>Fuel Oil<br>Gas | 18,912<br>2430<br>75,738 | 4100 | 2019 | **94 ca.** |
| Carbon<br>Fuel Oil<br>Gas | 18,912<br>2430<br>75,738 | 8760 | 2019 | **200 ca.** |
| Gas | 106,393 | 4100 | post 2025 | **91 ca.** |
| Gas | 106,393 | 8760 | post 2025 | **193 ca.** |

[1] Results obtained with calculating.

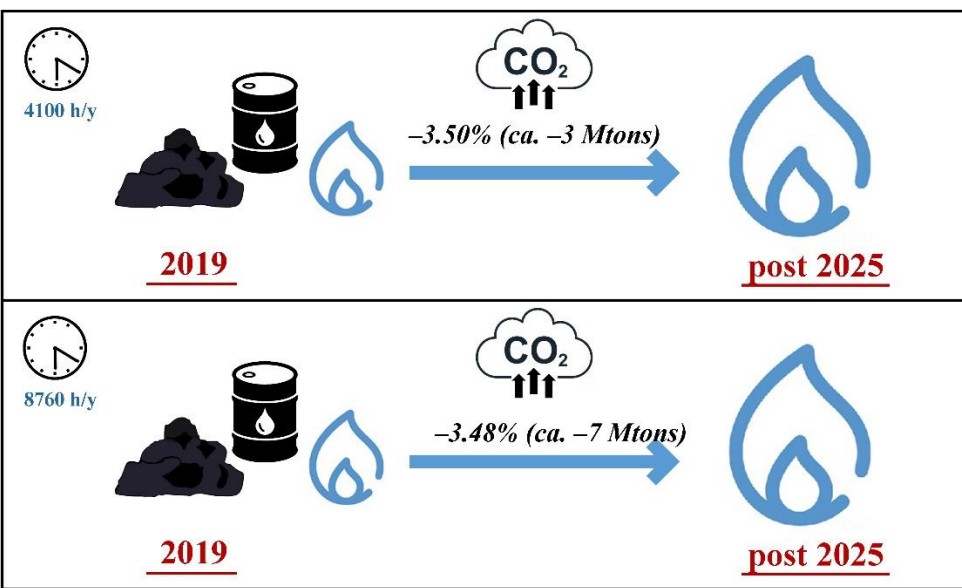

**Figure 7.** Theoretical reduction of $CO_2$ emissions.

The reliability of the above assessments, with reference to the value of $CO_2$ emissions referring to 2019, considering the real operation hours of the plants, is reflected in the recorded data relating to the quantity of $CO_2$ emitted in 2019 in Italy by the thermoelectric production sector, equal to at 93.8 Mt [25] (p. 62).

The applied methodology represents a simulation of the possible Italian emission scenario that can be reached in the coming years, starting from the data referring to the current layout of the energy sector and taking into consideration all the transformation projects that can be implemented. In this framework, the results of the processing carried

out, highlight a reduction in greenhouse gas emissions limited to only 3.5%, against the significant investments in progress for the transformation of the national energy sector.

To increase the limited environmental improvement resulting from the elaborations carried out, immediate optimization of the existing energy system could be implemented to obtain an immediate and early reduction of $CO_2$ emissions.

This optimization could be achieved, for example, by resorting to a more continuous operation of the existing thermoelectric plants, which allows to reduce the transitory phases (start-up and shut-down) that are highly polluting, using where necessary systems to store any energy surplus and guarantee in this way the balancing and security of the national grid, and carrying out technical/managerial interventions to increase the energy efficiency of the plants.

## 5. Conclusions

The targets of de-carbonization and fight to climate changes set at European level have led in Italy to the start of an important transformation process in the thermoelectric sector, with the complete interruption in the use of the most polluting fuels replaced by natural gas, to support an increasing production by plants that use renewable energy sources. In Italy, the total phase out of coal power generation by 2025, is reach by the end of the operations of the last 8 existing thermoelectric plants fueled by coal.

Considering all the projects for new plants or revamping of existing plants, that will be added to the plants currently in operation, the evaluations carried out highlight a reduction of $CO_2$ emissions in post-decarbonization layout limited to approximately 3.5%.

This result shows the need to implement further policies to reduce climate-altering emissions, also intervening on sectors different from the energy one, such as transport, civil heating, and agri-food, to ensure the achievement of the targets set by European and international agreements. An additional action can be realized optimizing the whole network of natural gas pipelines to reduce the contribute of the fugitive emissions to the global framework [26]. In relation to the energy sector, only by resorting to a prevalent use of energy produced by renewable sources will therefore be able to achieve a significant reduction of greenhouse gas emissions into the atmosphere, with the consequent pursuit of the targets set at European level. In this case, the resulting lower demand for energy produced with traditional plants that use fossil fuels, would not justify the high number of projects under construction for new gas plants, which appears disproportionate to the real needs of the coming years. These projects should be examined and evaluated in a coordinated and harmonized way and not individually, to characterize the areas and contexts that could suffer because of the use of renewable sources instead of coal plants and to identify the specific and necessary interventions to guarantee the effective safety and adequacy of the national electricity grid.

It should also be considered how an intensive use of methane generates problems both related to its supply, and to direct emissions into the atmosphere caused by losses in the transport, refueling phases, etc., since methane has a GWP (Global Warming Potential) of 28–36 times higher than the $CO_2$ [27]. This means that its climate-altering power is significantly greater than that of $CO_2$, therefore a greater contribution of methane emissions into the atmosphere could nullify the improvements deriving from the replacement of coal with gas in the electricity production sector. The alarm related to current methane emissions into the atmosphere was recently highlighted worldwide and is the subject of the Global Methane Pledge, a joint initiative of the United States and the European Union for the reduction of global methane emissions by 2030 by at least 30% compared to the levels recorded in 2020 [28].

The considerations on the intensive use of natural gas take on a further factor of concern considering the recent crisis linked to the lack of Russian gas supplies. The situation emerged in recent months has brought out even more clearly the need for a diversification of energy sources, trying to limit the use of fossil fuels, in Italy heavily

influenced by imports, and rationalizing the national energy system, considering the real needs of the country and the latest international political events.

Therefore, the Russo-Ukrainian war could give a decisive buster to the building of new energy plants fueled by renewable sources, in Italy wind and solar power could be increased; strengthening the projects funded under the NextGeneretionEU plan [29]. This transformation requires long times, therefore the current crisis in gas availability could immediately lead to a temporary return to the use of more polluting fossil fuels, both liquid and solid, with a possible slowdown in the achievement of goals related to the reduction of greenhouse gas emissions set for 2030.

**Author Contributions:** Conceptualization, C.C. and P.C.; methodology, C.C. and P.C.; formal analysis, P.C.; resources, C.C. and P.C.; data curation, C.C.; writing—original draft preparation, C.C.; writing—review and editing, C.C. and P.C.; visualization, C.C. and P.C.; supervision, C.C.; funding acquisition, A.F. and P.C. All authors have read and agreed to the published version of the manuscript.

**Funding:** This research received no external funding.

**Institutional Review Board Statement:** Not applicable.

**Informed Consent Statement:** Not applicable.

**Data Availability Statement:** All data are available from the corresponding author upon request.

**Conflicts of Interest:** The authors declare no conflict of interest.

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
