# Peer review of "The Italian Pathway for Energy Transition: From the Coal Phase Out to the Problems Related to Natural Gas"

_atmosphere, doi:10.3390/atmos13111872_

Round 1

Reviewer 1 Report

1.Considering the impact of the Russian Ukrainian war on European natural gas supply, can the established emission reduction target be achieved?The paper should strengthen the discussion in this regard.

2. The optimization of the existing energy system is also an effective way to reduce co2 emissions. Recent literature can be consulted, such as,10.1016/j.petsci.2022.09.025

Author Response

Point 1: Considering the impact of the Russian Ukrainian war on European natural gas supply, can the established emission reduction target be achieved? The paper should strengthen the discussion in this regard.

Response 1: Referring to the comment received, the paragraph 5 “Conclusions” was implemented with a consideration on this specific issue.

Point 2: The optimization of the existing energy system is also an effective way to reduce co2 emissions. Recent literature can be consulted, such as,10.1016/j.petsci.2022.09.025.

Response 2: Starting from the comment received the paper was improved with 2 further specific considerations; one on the paragraph 4 “Discussions” about the optimization of the thermoelectric plants, and one on the paragraph 5 “Conclusions” about the possibility of optimizing the natural gas pipelines network to reduce CO2 emissions, as explained in the interesting suggested paper.

Reviewer 2 Report

The authors should correct some typos, and give more emphasys to the new traced energy transition pathways.

1. Regarding the main question addressed by the research, the reviewer suggests the authors to better highlight the focus of the manuscript, especially in reference to the delicate "energy age" of the energy transition.

2. The introductory section must contain relevant and recent research or report papers published in relevant and high impact research Journal

3. The reviewer suggests to give a more scientific soundness, since the impression is of a good informative paper while reading it.

4. Emphasize the relevance and the originality of the manuscript  compared
with other published material.
5. I usually think that a flowchart is opportune to be included in a paper to well frame and present the conduction of the research.

6. The conclusions should report and repeat some important information and numbers, especially those inherent to the coal power plants to be turned off and those necessary to carry on the energy transition.

Author Response

General Comment: The authors should correct some typos, and give more emphasys to the new traced energy transition pathways.

Response to the General Comment: Regarding this comment some typos has been corrected, and the paper was improved introducing more consideration about energy transition pathways, mainly in the paragraphs n. 4 and n. 5, in agreement also with the other comments and suggestions proposed.

Point 1: Regarding the main question addressed by the research, the reviewer suggests the authors to better highlight the focus of the manuscript, especially in reference to the delicate "energy age" of the energy transition.

Response 1: Referring to the comment received, the paragraph n. 1 “Introduction” was implemented with a general framework of the issue and with an international contextualization of the topics covered in the paper.

Point 2: The introductory section must contain relevant and recent research or report papers published in relevant and high impact research Journal.

Response 2: Related to the comment received the paragraph n. 1 “Introduction” was improved with further specific bibliographic references of international relevance.

Point 3: The reviewer suggests to give a more scientific soundness, since the impression is of a good informative paper while reading it.

Response 3: The paper has been integrated mainly in the paragraphs n. 4 and n. 5 to highlight the results obtained with the elaborations carried out. As suggested also a flowchart has been added to represents the methodological approach adopted.

Point 4: Emphasize the relevance and the originality of the manuscript compared with other published material.

Response 4: The originality of the manuscript was to recover, compare and re-elaborate data and information, about Italian energy sector, available on institutional sites, to obtain, with a specific calculation system developed by the authors, an estimate of the reduction in greenhouse gas emissions deriving from the transformation of the electricity sector underway in Italy.

The use that was made with the input data and the result obtained are not comparable with any other work because they are characteristic and specific of the context represented in the paper.

Point 5: I usually think that a flowchart is opportune to be included in a paper to well frame and present the conduction of the research.

Response 5: As suggested, a flowchart has been added in the paragraph n. 4, to represents the methodological approach adopted.

Point 6: The conclusions should report and repeat some important information and numbers, especially those inherent to the coal power plants to be turned off and those necessary to carry on the energy transition.

Response 6: Referring to the comment received, the paragraph n. 5 “Conclusions” was implemented with the information requested as suggested and with further information on the energy transition issue.

Reviewer 3 Report

The manuscript submitted by the authors like a magazine article. It is only describing some of the laws passed by the European Commission Communication. There are no sufficient comparison or inference with the real time situations. The authors have to incorporate more discussion on the current scenario.

Author Response

General Comment: The manuscript submitted by the authors like a magazine article. It is only describing some of the laws passed by the European Commission Communication. There are no sufficient comparison or inference with the real time situations. The authors have to incorporate more discussion on the current scenario.

Response to the General Comment: The input data taken as a starting point for the elaborations carried out in the paper are available and published on institutional Italian websites. The scientific originality of the manuscript derives from the processing of such data to obtain, with a specific calculation system developed by the authors, an estimation of the reduction in greenhouse gas emissions deriving from the transformation of the electricity sector underway in Italy, according with EU industrial laws and international objectives.

Considering this reference context, this manuscript was specifically submitted for the special issue of the journal “Atmosphere” titled "Industrial Air Pollution: Emission, Management and Policy”.

Referring to the comments received the manuscript has been integrated and expanded in several parts, mainly in the paragraphs n. 4 and n. 5 to highlight the results obtained with the elaborations carried out. Also, the introduction has been updated to better define the international context of reference.

Round 2

Reviewer 3 Report

The authors have given sufficient justification for the comments raised. Regarding the refinement, I would like to draw attention to the following: in the work at the stage. A minor revision is required. The article is recommended for publication in the journal after revision without re-reviewing.

1. Improve the quality of the figure 2 and 4 for better visibility.

2. Mark north arrow tag in map.

3. References must be avoided in the conclusion.

4. Authors can incorporate some historical case studies for better understanding of readers.

Author Response

General Comment: The authors have given sufficient justification for the comments raised. Regarding the refinement, I would like to draw attention to the following: in the work at the stage. A minor revision is required. The article is recommended for publication in the journal after revision without re-reviewing.

Response to the General Comment: no respons required.

Point 1: Improve the quality of the figure 2 and 4 for better visibility.

Response 1: the quality figure 2 was improved, the quality of figure 4 is the maximum available on the original source.

Point 2: Mark north arrow tag in map.

Response 2: the mark north arrow was integrated in the figure 5.

Point 3: References must be avoided in the conclusion.

Response 3: in the opinion of the authors some references reported in the conclusion are usefully for a better framework of the element of the paragraph; some references have however been removed

Point 4: Authors can incorporate some historical case studies for better understanding of readers.

Response 4: in the opinion of the authors considering the originality of the element showed in the paper is not possible insert relevant and significative historical case studies.